# The Significant Potential of Simonkolleite Powder for Deep Wound Healing under a Moist Environment: In Vivo Histological Evaluation Using a Rat Model

**DOI:** 10.3390/bioengineering10030375

**Published:** 2023-03-19

**Authors:** Osamu Yamamoto, Miki Nagashima, Yoshimi Nakata, Etsuro Udagawa

**Affiliations:** 1Graduate School of Science and Engineering, Yamagata University, 4-3-16 Jonan, Yonezawa 992-8510, Japan; 2Research Laboratories, JFE Mineral & Alloy Co., Ltd., 1 Niihama-cho, Chuo-ku, Chiba 260-0826, Japan

**Keywords:** ceramics powder, bioactive ion, regenerative skin, wound healing

## Abstract

In the present work, simonkolleite powder consisting of Zn_5_(OH)_8_Cl_2_·H_2_O composition was proposed as a new candidate material for the healing of deep wounds in a moist environment. The powder was synthesized using a solution process and evaluated for wound-healing effects in rats. The pH value of physiological saline at 37 °C using the simonkolleite powder was 7.27, which was the optimal pH value for keratinocyte and fibroblast proliferation (range: 7.2–8.3). The amount of Zn^2+^ ions sustainably released from simonkolleite powder into physiological saline was 404 mmol/L below cytotoxic ion concentrations (<500 mmol/L), and the rhombohedral simonkolleite was accordingly converted to monoclinic Zn_5_(OH)_10_·2H_2_O. To evaluate the wound-healing effect of simonkolleite powder, the powder was applied to a full-thickness surgical wound reaching the subcutaneous tissue in the rat’s abdomen. The histological analysis of the skin tissues collected after 1, 2, and 4 weeks found that angiogenesis, collagen deposition, and maturation were notedly accelerated due to the Zn^2+^ ions released from simonkolleite powder. The simonkolleite regenerated collagen close to autologous skin tissue after 4 weeks. The hair follicles, one of the skin appendages, were observed on the regenerative skin in the simonkolleite group at 4 weeks but not in the control group. Therefore, simonkolleite was hypothesized to stimulate the early regeneration of skin tissue in a moist environment, compared with commercial wound dressing material. These results suggested that simonkolleite could offer great potential as new wound dressing material.

## 1. Introduction

Skin acts as a biological defense against toxic bacteria and harmful wavelengths of sunlight and is thus a biological tissue of vital importance for health. In order to heal wounds with full-thickness skin defects in dermatology and plastic surgery, skin transplantation using autograft, allograft, and xenograft has been used in clinical treatment. However, skin transplantation has clinical problems such as invasive surgical procedures and intense immune rejection, etc. [1,2]. For severe skin damage such as deep wounds, suturing the skin is the first choice, but suturing is also accompanied by strong contracture over a long period. Especially wound therapy, which would not cause scarring and contracture, has been essential to patients and bedridden elderly with severe skin damage. Wound healing in a moist environment using wound dressing materials is well-known to be effective in healing, facilitating cell migration from the wound edge, but wound healing time depends on the depth and width of the skin wound. Although many wound therapies are available, challenges remain in treating problematic injuries such as deep wounds and excess exudate. Concerning problematic wounds, promoting angiogenesis and collagen production followed by re-epithelialization is essential to regenerate the normal tissue in the damaged area because blood vessels transport nutrients, minerals, and oxygen in the body to the wound site [3,4,5].

Over the past few decades, various inorganic and organic biomaterials have been studied and clinically applied for their healing effects on skin wounds. Organic wound dressing materials used in wet wound therapy have been proposed for skin repair, including mats, films, sponges, hydrocolloids, hydrogels, and many others [6,7,8,9,10,11]. Focusing on inorganic materials, bioactive glasses have been reported to have wound healing [12,13,14,15,16]. Most wound-healing materials have shown good healing for superficial wounds extending from the epidermis to the subcutaneous tissue. For ceramics excluding silicate glasses, a few reports on their deep wound-healing effects were published a few years ago [17,18]. Among the reports on wound-healing materials, many researchers also focused on bioactive ions such as Zn^2+^, Mg^2+^, and Si^4+^ ions, etc., which have been known to be extremely effective for tissue regeneration and repair [19,20,21,22,23,24,25]. The Zn^2+^ ions in these bioactive ions have attracted particular attention. The reason is that the Zn^2+^ ions are a critical component of many proteins that play an important role in physiological processes [26,27]. Specifically, matrix metalloproteinases (MMPs) are a class of zinc-dependent endopeptidases involved in the wound-healing process [28,29,30]. For example, MMP-6 is a member of the MMP family that promotes angiogenesis and cell growth through the hydrolysis of collagen molecules. Therefore, Zn^2+^ ions should ultimately be effective in completing the wound healing process.

Simonkolleite with Zn_5_(OH)_8_Cl_2_·H_2_O composition is known as an intermediate in the process of synthesizing the semiconductor ZnO. In the simonkolleite, 1/4 of the octahedrally coordinated zinc within a layer is replaced by pairs of tetrahedra with the Zn_octahedra_/Zn_tetrahedra_ ratio of 1.5, resulting in trigonal layer symmetries for the chloride. The trigonal layer in the simonkolleite phase is anticipated to be capable of releasing Zn^2+^ ions in water due to the presence of Cl^−^ ions. Hence, we focus on simonkolleite as a new candidate material for wound treatment, because of the possibility of sustained release of Zn^2+^ ions. However, no effect of simonkolleite on wound healing is clarified; there are no reports. This study aims to prepare simonkolleite powder and evaluate the wound-healing effects of simonkolleite on full-thickness skin defects, i.e., deep wounds, using a rat model.

## 2. Materials and Method

### 2.1. Synthesis of Simonkolleite Powder Using a Solution Process

Ammonium chloride (NH_4_Cl, purity 99.8%, Nacalai Tesque, Kyoto, Japan), zinc chloride (ZnCl_2,_ purity 98%, Nacalai Tesque, Kyoto, Japan), and sodium hydroxide (NaOH, purity 97%, Nacalai Tesque, Kyoto, Japan) were used as raw materials. In total, 500 mL of ammonium chloride aqueous solution with a concentration of 0.2 mol/L and 1000 mL of zinc chloride aqueous solution with a concentration of 0.5 mol/L were prepared. The zinc chloride aqueous solution was then added dropwise to the ammonium chloride aqueous solution at a rate of 500 mL/h, and the pH of the mixed solution was maintained at 6.5 by adding a 30 mass% sodium hydroxide aqueous solution. After the resulting suspension was stirred at 26 °C for 16 h, solid–liquid separation was performed using suction filtration. The synthesized powder was washed several times with 100 mL of distilled water and dried at 40 °C under vacuum conditions to obtain simonkolleite powder. In a ball milling, a 500 mL capacity pot mill (Irie shokai, Tokyo, Japan) made of propylene was used, and the simonkolleite powder was mixed to a mass ratio of one-tenth to zirconia balls with a diameter of 10 mm (Nikkato Co., Osaka, Japan). The simonkolleite powder was ground in a ball milling for 72 h at the rotation speed of 50 rpm. Then, the ground simonkolleite powder was applied to full-thickness skin defects in rats.

### 2.2. Powder Characterizations

The synthesized simonkolleite powder before and after ball milling was analyzed using field emission scanning electron microscopy (SEM, Sigma 500VP, ZEISS, Oberkochen, Germany), to observe the shape of the powder, and powder X-ray diffraction meter (XRD, UltimaIV, Rigaku, Tokyo, Japan). The XRD measurement conditions were as follows: voltage: 40 kV, current: 40 mA, scanning speed: 2/min, and step width: 0.02. The XRD diffraction peaks were corrected using the silicon standard method for normalization.

### 2.3. Zn^2+^ Ions Releasing Test and pH Value

It is important that the solution used for the Zn^2+^ ions release test resembles the osmotic pressure for cells in the body. Physiological saline (0.9% NaCl solution, medical grade, Otsuka Pharmaceutical Co., Tokushima, Japan) was used for the ion release test, due to a similar osmolarity to the body.

In order to measure the concentration of Zn^2+^ ions released from simonkolleite powder, 1.5 g of simonkolleite powder was dispersed in 5.0 mL of physiological saline. The physiological saline containing Zn^2+^ ions was collected every 24 h for 7 days. The Zn^2+^ ions concentration and pH value in the collected saline were measured using inductively coupled plasma-mass spectroscopy (ICP-MS, ELAN DRCII, Perkin Elmer Japan Co., Ltd., Kanagawa, Japan) and a pH meter (HI 2020-1; HANNA Instruments Japan, Chiba, Japan), respectively. The crystal structure of simonkolleite after the ion release test was identified using XRD.

### 2.4. In Vivo Evaluation of Simonkolleite

#### 2.4.1. Animal Experiment

##### Preparation for Animal Surgery

Prior to animal surgery, simonkolleite powder was sterilized using irradiation with ultraviolet light at a wavelength of 240 nm for 48 h to prevent bacterial infection. All surgical instruments such as scalpels, tweezers, and forceps were sterilized using an autoclave (LBS-245; Tomy Seiko Co., Ltd., Tokyo, Japan) at 121 °C for 20 min.

##### Animal Models

The protocols for the animal experiment described in this study were approved by the animal committee of Yamagata University (Yamagata, Japan). Six rats (n = 6) (Male, Jcl: SD, CLEA Japan, Inc., Tokyo, Japan), weighing 345 ± 42 g, were used in the study for each healing time in each group. Rats were spontaneously inhaled through the nose using 3% sevoflurane (Maruishi Pharmaceutical Co., Ltd., Osaka, Japan) as an inhaled anesthetic. Subsequently, the rats were sedated with an intramuscular injection of 10 mg/kg xylazine hydrochloride (Sedeluck, Nippon Zenyaku Kogyo Co., Ltd., Fukushima, Japan) as a muscle relaxant. Abdominal hair was completely removed with an electric clipper and hair removal cream. After disinfection at the epilation site with 7% povidone-iodine, 2% lidocaine hydrochloride containing 1:80,000 epinephrine (Xylocaine Poly Amp 2%, Fujisawa Pharmaceutical Co., Ltd., Osaka, Japan) was administered subcutaneously near the portion creating skin defects. Four skin defects were created on the left and right ventral sides using a 10 mm diameter trephine bar. As shown in Figure 1, the skin defect was covered with commercial wound dressing material (Duoactive ET^®^, Convatec Inc., Bridgewater, NJ, USA) in the control group. In contrast, the simonkolleite groups were first applied with 0.005 g of simonkolleite powder and then covered with the same commercial wound dressing material. The evaluation was conducted at 1-, 2-, and 4-week intervals.

#### 2.4.2. Morphometric Analysis

A digital photograph of the wounds was taken after 0, 1, 2, and 4 weeks of healing time. The residual area of the wounds based on the photograph was extracted from the captured images using an image-processing program (Image J, National Institutes of Health, Bethesda, MD, USA). The percentage of residual wound area was calculated using the following Equation (1):(1)$$\mathrm{Percentage}\;\mathrm{of}\;\mathrm{residual}\;\mathrm{wound}\;\mathrm{area}=\frac{W_{0}}{W_{t}}\times 100$$
where W_0_ and W_t_ represent the initial wound area and the wound area measured at 1, 2, and 4 weeks, respectively.

#### 2.4.3. Histological Evaluation

After 1, 2, and 4 weeks, the rats were euthanized using an intraperitoneal overdose of anesthetic (Pentobarbital sodium salt, Nacalai tesque, Inc., Kyoto, Japan). Skin tissue, including wounds, was collected from rats for histological evaluation. The excised tissues were fixed with 20% paraformaldehyde-containing phosphate buffer for 48 h. The fixed tissues were then embedded in a paraffin wax block. The specimens were sectioned into 6–8 µm thicknesses from the block. The sections were then stained using hematoxylin and eosin staining (HE staining: Muto Pure Chemicals, Co., Ltd., Tokyo, Japan) and Masson’s trichrome staining (MT staining: Muto Pure Chemicals, Co., Ltd., Tokyo, Japan). Cells and collagen generated during wound healing were observed on the stained sections with an optical microscope (BX53, OLYMPUS, Tokyo, Japan).

#### 2.4.4. Blood Vessels Counting

Next, 20-fold magnification photographs (2448 × 1980 pixels) were taken on the regenerated tissue using the HE-stained sections. The area of each photograph was converted from pixel to mm units using the Image J program (NIH, Bethesda, MD, USA). Five arbitrary areas were selected from the tissue regenerated in the wound of each sample. The blood vessels within the regenerated tissue were counted based on clear observation of the vessel lumen. Individual counts were repeated 3 times. The number of blood vessels per 1 mm^2^ was calculated using the following Equation (2):(2)$$\mathrm{Number}\;\mathrm{of}\;\mathrm{new}\;\mathrm{blood}\;\mathrm{vessels}\;[\mathrm{units}/{\mathrm{mm}}^{2}]=\frac{BV}{A}$$

In the above equation, BV and A represent the number of blood vessels and the area of each image, respectively.

### 2.5. Statistical Analyses

For each parameter, a one-way Welch’s t-test was selected based on its statistical variance. Percent reduction in wound size and the number of vessels were evaluated based on statistical significance (*p*-value < 0.05) at the 5% significance level. The error bar indicates the standard deviation (SD), with the upper and lower limits expressed as ±.

## 3. Results

### 3.1. Characterization of Simonkolleite

Among the elements in the human body, Zn^2+^ ions have a big impact on tissue growth and disease progression. In our experiments, simonkolleite powder consisting of Zn^2+^, OH^−^, and Cl^−^ was prepared using the solution process. The simonkolleite powder obtained was ground with ball milling to obtain uniform fine particles. The powders before and after milling were analyzed using XRD and SEM.

Figure 2 shows the results of XRD measurements on the simonkolleite powder. The XRD diffraction peaks corresponding to simonkolleite (ICDD: No.01-074-3156) were detected in the powder before and after milling, without other crystal phases. The crystallinity before milling was slightly lower than that after milling, probably due to slight stresses on the crystal structure caused by milling. The SEM micrographs of the simonkolleite powder are shown in Figure 3.

Before milling (Figure 3a), the powder was aggregated with a diameter of 20 to 40 mm, containing small scale-like particles with a diameter of 1 µm and thickness of 0.1 µm. Close observation revealed that the tips of the scale-like particles were sharp like blades. After milling, the powder was observed as aggregates or dispersion of spherical particles with a diameter below 0.1 mm, as shown in Figure 3b.

The concentration of Zn^2+^ ions released from simonkolleite is shown as a function of soaking time in Figure 4. The concentration of Zn^2+^ ions was at a maximum value (404 mmol/L) on day 6, and then sustainable release of Zn^2+^ ions during 1–6 days was observed.

Figure 5 shows the XRD patterns of simonkolleite before and after soaking in physiological saline for 7 days. The XRD diffraction peaks corresponding to Zn_5_(OH)_10_·2H_2_O and simonkolleite were observed by immersing single-phase simonkolleite in the saline solution for 7 days; that is, it resulted that Zn_5_(OH)_10_·2H_2_O was deposited by releasing Zn^2+^ and Cl^−^ ions from simonkolleite. The (002) diffraction peak around 2q ≈ 11.5° in the simonkolleite shifted to the high-angle side with the deposition of Zn_5_(OH)_10_·2H_2_O.

In this study, the size of the wound created was 10 mm in diameter and 3 mm in depth. Based on the wound size and the concentration of Zn^2+^ ions below cytotoxicity, the appropriate powder concentration was calculated, and 0.005 g of simonkolleite powder was applied at each wound site.

### 3.2. The Wound Healing Effect of Simonkolleite

#### 3.2.1. Macroscopic Changes in the Residual Wound Area

Figure 6 shows the gross photos of the wound after each healing time. Wound healing was grossly similar between the control group and the simonkolleite group. However, when the wounds were closely observed at 1 week, the uniform red-white tissue recognized as granulation tissue was observed in the simonkolleite group, but not in the control group. At 2 weeks, the simonkolleite group seemed to have a pronounced epithelialization, and the wound area was almost covered with new thin epithelium. On the other hand, in the control group, there was no epithelial formation in one part of the wound site. At 4 weeks, there were traces of healing, but no residual wound was observed. From those gross observations, it appeared that the residual wound of the simonkolleite group with the healing period tended to be a little smaller than those of the control group. Table 1 summarizes the residual wound area at each healing time as determined using the image J processing program. The residual wound area decreased with increasing healing time in both groups, and there was no significant difference (*p* < 0.05) between the groups. In the case that the histological state of wound healing is uncertain in the gross observation, as shown in Figure 6, it is important to infer the healing process from the staining of the healing tissue.

#### 3.2.2. Histological Evaluation

To investigate the healing effect of simonkolleite powder in a moist environment, commercially available wound dressing (Duo active ET^®^) was used and compared with and without the application of simonkolleite powder. The histological evaluation was performed on the wound tissue collected after each healing time using HE and MT staining. As a reference for these stained tissues, undamaged autologous skin tissues were also subjected to HE and MT staining.

Figure 7 shows the images of HE- and MT-stained autologous skin tissue. As seen in the low magnification image of the HE staining, stratum corneum, granular layer, basal layer, and hair follicle were observed in the epidermis, while fibroblasts in the subcutaneous tissue and dermis were also seen in the high-magnification image. Blue thick collagen fibers and light red elastic fibers could be distinguished from the high magnification of the MT-stained skin.

High magnification of HE- and MT-stained skin in the wound healing site at each healing time are shown in Figure 8. In comparison with the autologous skin tissue described above, at the wound healing site after 1 week (Figure 8a), inflammatory cell infiltration such as granulocytes and lymphocytes, angiogenesis, and collagen synthesis were observed in both the control and simonkolleite groups. The formation of collagen fiber seems to be of a low density in the control group compared with the simonkolleite group. The remarkable formation of a myxoid-type connective tissue was observed in the control group. At 2 weeks (Figure 8b), the inflammatory cell infiltration seen at 1 week was still observed in the control and simonkolleite groups, but the inflammation seems to be suppressed after 2 weeks rather than 1 week. Thick dense collagen production was noted in the simonkolleite group at this time point, and these collagen fibers resembled autologous skin tissue. In the control group, the collagen deposition at the same time point was low density and the collagen fibers were fine. At 4 weeks (Figure 8c), the control group showed the same inflammatory cell infiltration as at 2 weeks. In contrast, fibroblasts and a few lymphocytes were observed in the simonkolleite group, indicating the absence of inflammatory cell infiltration. Collagen density in the control group was higher than at 2 weeks, but the thickness of its fibers remained the same. In the simonkolleite group, the thick collagen fibers seen in autologous skin tissue were observed, suggesting that skin remodeling was in progress. From these stained images, it was found that no simonkolleite powder remained in the skin tissue regenerated during the healing time.

An overall view of the HE-stained skin after 4 weeks and the HE images focusing on the regenerated skin area are shown in Figure 9. In the control and simonkolleite groups, subcutaneous muscle tissue was regenerated from the limbus of the initial wound toward the center of the wound site but not over the entire defect site. Epithelium consisting of epidermis and dermis was formed in both groups. In detailed observations, however, the regeneration of skin appendages such as hair follicles identical to the autologous skin tissue (Figure 7) was observed in the simonkolleite group.

Figure 10 shows the number of new blood vessels at each healing time. In the simonkolleite group, the number of new blood vessels showed a maximum value at 1 week and decreased with increasing healing time. In comparison with the control, the number of blood vessels in the simonkolleite group was significantly higher at 1 week and significantly lower at 4 weeks. That is, angiogenesis in the simonkolleite group was indicated to be faster and more prolific than in the control group.

## 4. Discussion

Simonkolleite powder was synthesized from an aqueous solution containing NH_4_Cl, ZnCl_2,_ and NaOH. The reaction Equation (3) for synthesizing the powder can be written as follows.
5ZnCl_2_ + NH_4_Cl + 8NaOH + nH_2_O → Zn_5_(OH)_8_Cl_2_·H_2_O + 8NaCl + NH_3_ + HCl + (n−1)H_2_O (3)

Zn_5_(OH)_10_·2H_2_O was deposited with the release of Zn^2+^ and Cl^−^ ions from simonkolleite and the (002) diffraction peak around 2q ≈ 11.5° shifted to the high-angle side. The general formula of simonkolleite can be described as (Zn^2+^)_×_(OH^−^)_2x-my_(A^m−^)_y_·nH_2_O (A^m−^ = Cl^−^, NO_3_^−^, etc.), which has a layered structure. Simonkolleite has rhombohedral structure. The octahedrally coordinated zinc within a layer is replaced by pairs of tetrahedra. Neutral H_2_O is intercalated between the layers, and the layers are stacked [31]. In contrast, the structure of Zn_5_(OH)_10_·2H_2_O is monoclinic. The tetrahedral units within the layers are identical to that of Zn_5_(OH)_8_(NO_3_)_2_·2H_2_O [32]. The stacking of layers in the Zn_5_(OH)_10_·2H_2_O is similar to that in simonkolleite, which also contains H_2_O in neutral layers. As drawn in Figure 11, the interlayer distance in Zn_5_ (OH)_10_·2H_2_O has been known to be slightly shorter than that in simonkolleite [33,34]. Therefore, the shift in the (002) diffraction peak can be attributed to the reduction in the interlayer distance of the H_2_O intercalated in neutral layers. The compositional change due to the release of Zn^2+^ and Cl^−^ ions from simonkolleite can be expressed by the following Equation (4):Zn_5_(OH)_8_Cl_2_·H_2_O → 4/5Zn_5_(OH)_10_·2H_2_O + Zn^2+^ + 2Cl^−^ + nH_2_O(4)

Zn^2+^ ions were released sustainably from the simonkolleite powder for 6 days, with a maximum value of 404 mmol/L. Lansdown et al. [28] have reported that the inhibitory concentration of Zn^2+^ ions on cell proliferation is greater than 500 µmol/L. Thus, simonkolleite powder is presumed to be unlikely to cause cytotoxicity.

The pH value of physiological saline at 37 °C using the simonkolleite powder before and after milling was 7.49 and 7.27, respectively. Sharpe et al. [35] have reported that the optimal pH value for keratinocyte and fibroblast proliferation was in the range of 7.2–8.3. The pH value in a living organism is in the steady-state range of 7.3–7.4. Even if simonkolleite powder is applied to the wound, the resulting pH value can be estimated to have no adverse effect on wound healing. Since materials such as powder with sharp edges to the wound might be supposed to cause inflammation with physical irritation, it should be readily presumed that a spherical shape with few sharp edges is suitable for the shape of simonkolleite powder. Accordingly, the simonkolleite powder after milling was used as the wound-healing material in this work.

Since the integument provides the first barrier against invading microbes and pathogens, restoration of tissue integrity and homeostasis following injury to the skin is of vital importance [36]. Wound healing is a complex process involving a number of coordinated events, including matrix generation, angiogenesis, inflammatory cell expression, cell migration, and cell proliferation that contribute to skin regeneration [37]. The wound healing process consists of four key phases such as hemostasis, inflammation, proliferation, and a remodeling phase, which overlaps each phase with healing progresses in vivo [28,38]. These phases can be characterized by different cell types. For instance, an infiltrate of various inflammatory cells, such as neutrophils and macrophages, is observed during the inflammatory phase. In the proliferative phase, granulation tissue is generated by the accumulation of fibroblasts and vascular endothelial cells. As a result, collagen and new blood vessels are produced in the tissue. Re-epithelialization is caused by the migration of epithelium cells from the wound edge. Finally, the maturation of collagen and the reduction in the blood vessel are progressed to the remodeling phase. That is, during the early stages of wound healing, a significant increase in the number of blood vessels is expected to accelerate wound healing. At 1 week, the simonkolleite group showed granulation tissue formation and a significant increase in new blood vessels. Considering the wound-healing process, the tissue of the simonkolleite group would be presumed as a proliferative phase. In contrast, the control group would be presumed as an inflammatory phase since many inflammatory cell infiltrations such as lymphocytes and neutrophils were observed in the wound area at the same time. At 2 weeks, in the simonkolleite group, a systemic arrangement of the mature collagen bundle was found in the covered wound area, although inflammatory cell infiltration remained. The healing stage might be an intermediate stage migrating from proliferation to the remodeling phase. On the other hand, in the control group, inadequate collagen maturation and residual inflammatory cell infiltration were observed at the same time, which suggests an intermediate stage between the inflammatory and proliferative phases. At 4 weeks, the simonkolleite group showed regeneration of hair follicles resembling autologous skin tissue, which was not seen in the control group. That is, the simonkolleite group indicates being in the remodeling phase at 4 weeks. The advance in the wound healing process based on the above discussion is also supported by a decrease in the number of new blood vessels with the healing time. These results indicate that the wound healing of the simonkolleite group strongly suggests healing faster than the control group. Zn^2+^ ions have antioxidant and anti-inflammatory activities [39,40]. Furthermore, zinc ions stimulate angiogenesis by activating growth factors such as VEGF, Insulin Growth Factor 1 (IGF-1), and Transforming Growth Factor beta-1 (TGF-β1) [41]. From these reports, it is supposed that the Zn^2+^ ions released from simonkolleite would act on MMPs being the zinc-dependent endopeptidase and the activation of growth factors. In view of the above discussion, the pronounced healing effect of simonkolleite can be proposed due to the promotion of angiogenesis and cell proliferation through the dissolution of collagen molecules at the early stages of wound healing.

## 5. Conclusions

Crystalline simonkolleite powder was synthesized using a solution process. Zn^2+^ ions were released sustainably from the powder, which had a maximum value of 404 mmol/L. Zn_5_(OH)_10_·2H_2_O was deposited with the release of Zn^2+^ ions from simonkolleite. Compared with the commercial wound dressing, the simonkolleite powder significantly promoted collagen formation and angiogenesis in wounds due to the Zn^2+^ ions released from simonkolleite. Simonkolleite resulted in the acceleration of wound healing in a moist environment. From these results, simonkolleite can offer great potential as new wound dressing material in the future.

## Figures and Tables

**Figure 1 bioengineering-10-00375-f001:**
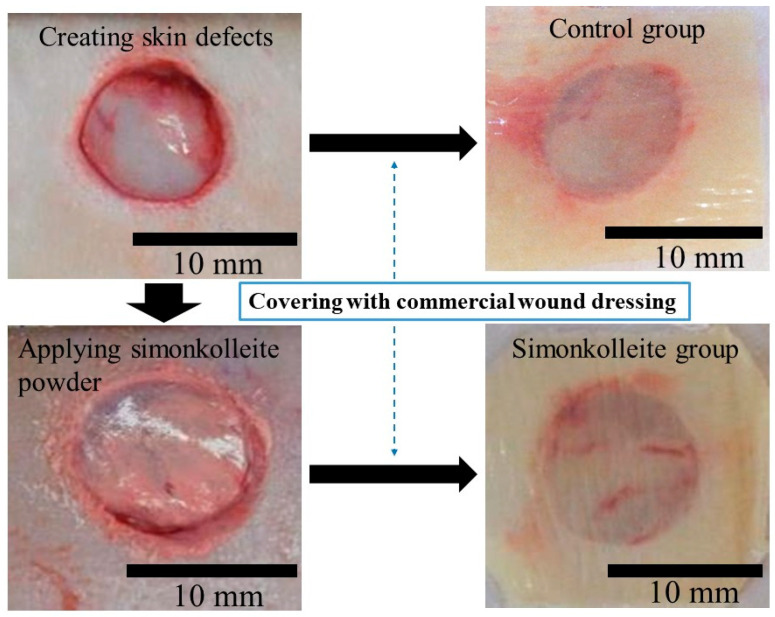
The application status of simonkolleite powder and wound dressings in each group.

**Figure 2 bioengineering-10-00375-f002:**
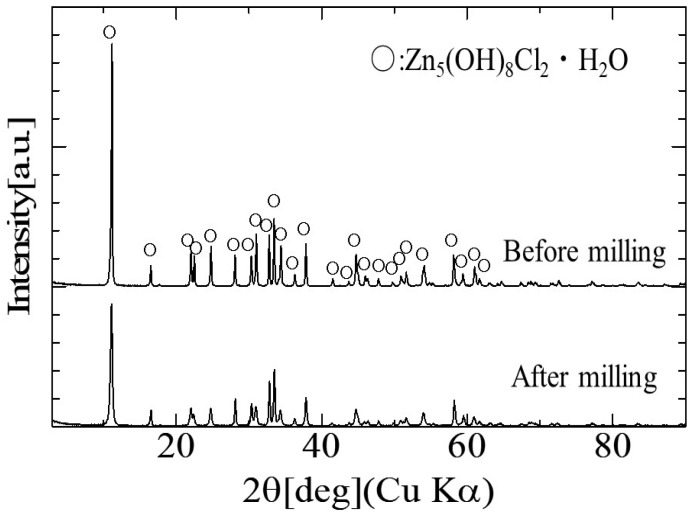
The X-ray diffraction patterns of simonkolleite before and after ball milling for 72 h.

**Figure 3 bioengineering-10-00375-f003:**
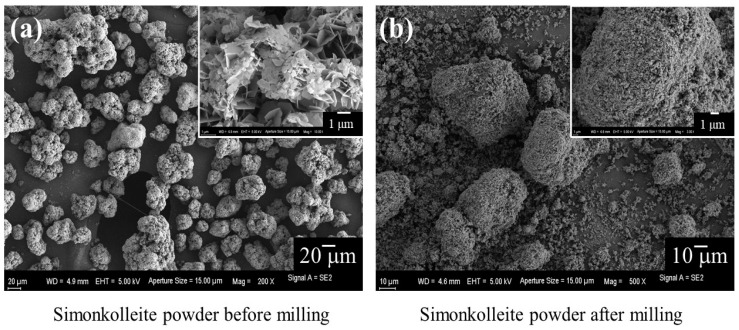
The SEM micrographs of the simonkolleite powder before (**a**) and after (**b**) milling.

**Figure 4 bioengineering-10-00375-f004:**
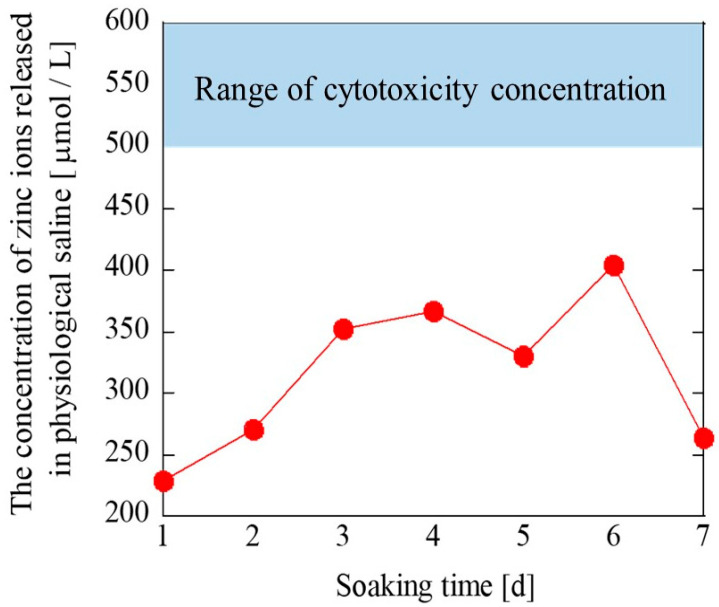
The release profile of Zn^2+^ ions released from the milled simonkolleite in physiological saline.

**Figure 5 bioengineering-10-00375-f005:**
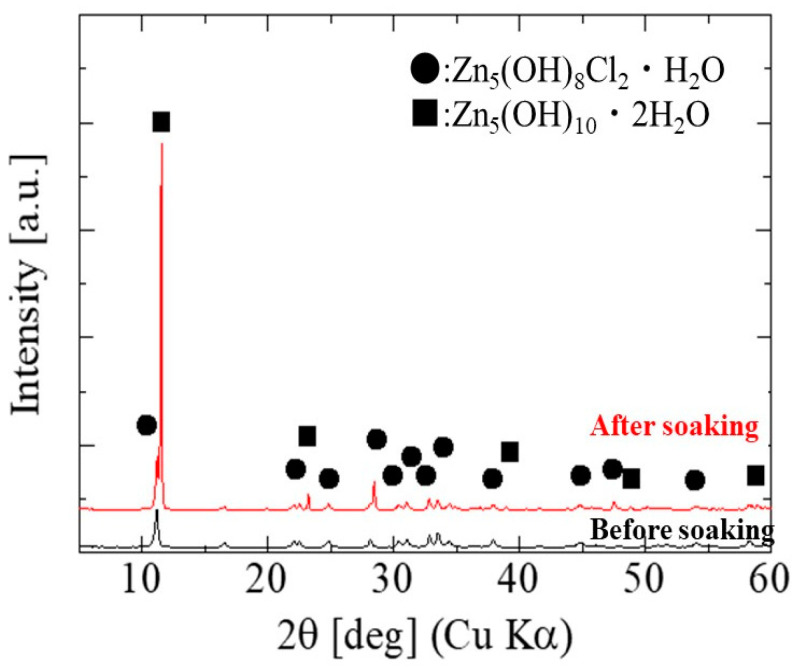
The X-ray diffraction patterns of simonkolleite before and after soaking in physiological saline for 7 days.

**Figure 6 bioengineering-10-00375-f006:**
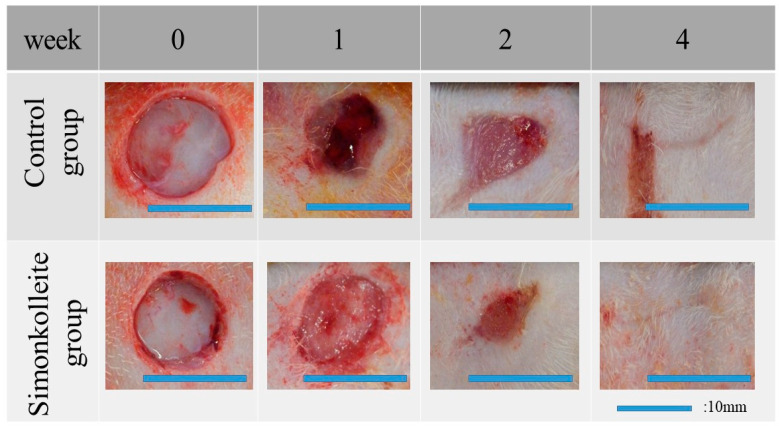
Gross observations of the wounds in the simonkolleite and control groups at each time point.

**Figure 7 bioengineering-10-00375-f007:**
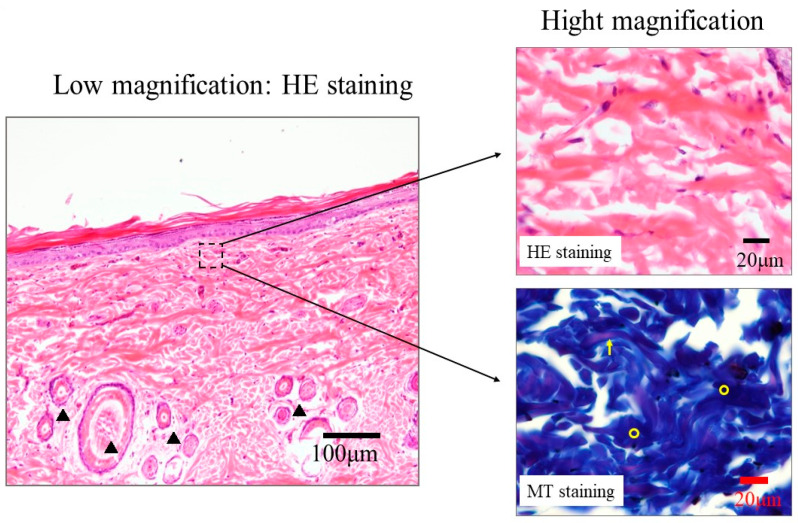
The images of HE- and MT-stained autologous skin tissue of a rat. The symbols of ○, ↑, and ▲ mean collagen fiber, elastic fibers, and hair follicle, respectively.

**Figure 8 bioengineering-10-00375-f008:**
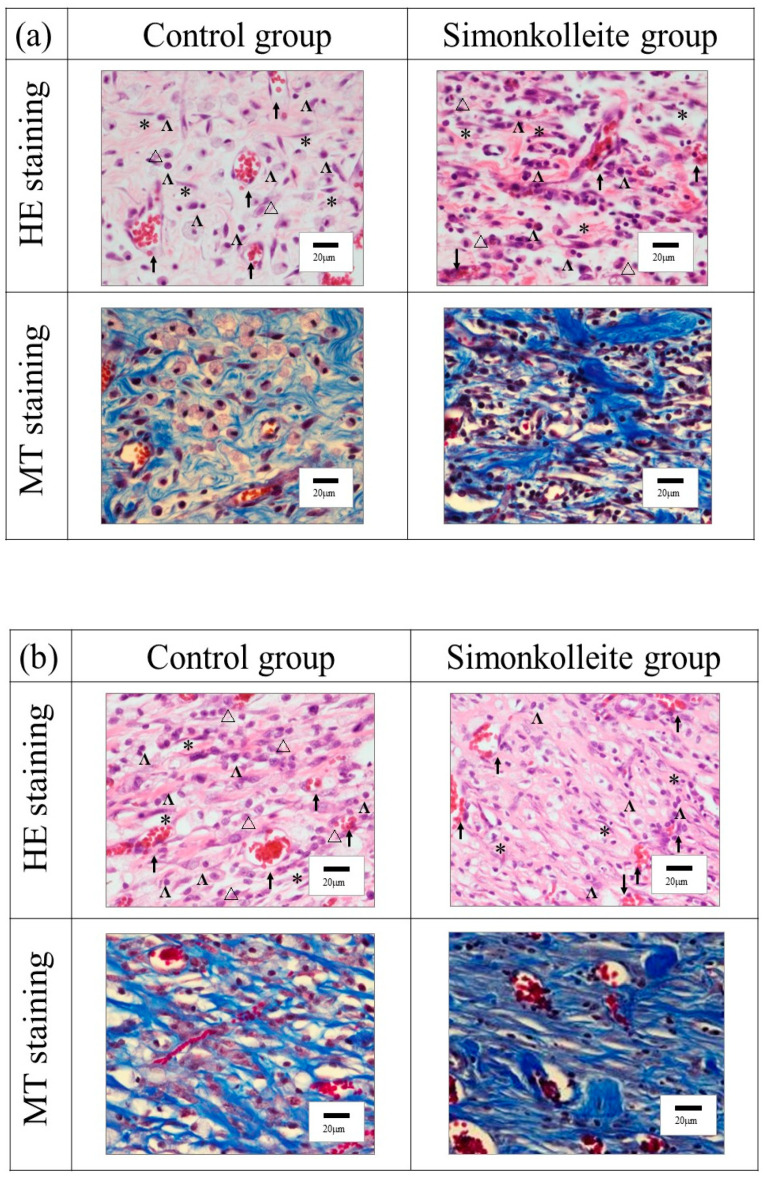
The high magnification images of HE- and MT-stained skin in the wound healing site after (**a**) 1 week, (**b**) 2 weeks, and (**c**) 4 weeks of healing time. ↑: Angiogenesis; *: Fibroblast; Δ: Granulocyte; Λ: Lymphocyte.

**Figure 9 bioengineering-10-00375-f009:**
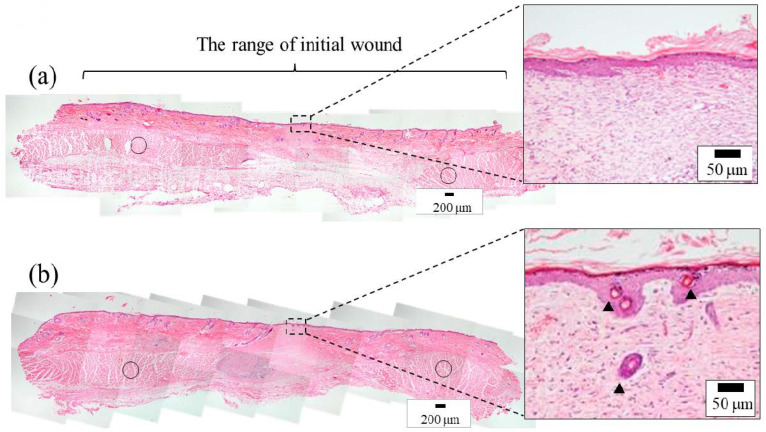
An overall view of the HE-stained skin after 4 weeks and the HE images focusing on the regenerated skin area: (**a**) control group and (**b**) simonkolleite group. The symbols of ○ and ▲ mean muscle tissue and hair follicle, respectively.

**Figure 10 bioengineering-10-00375-f010:**
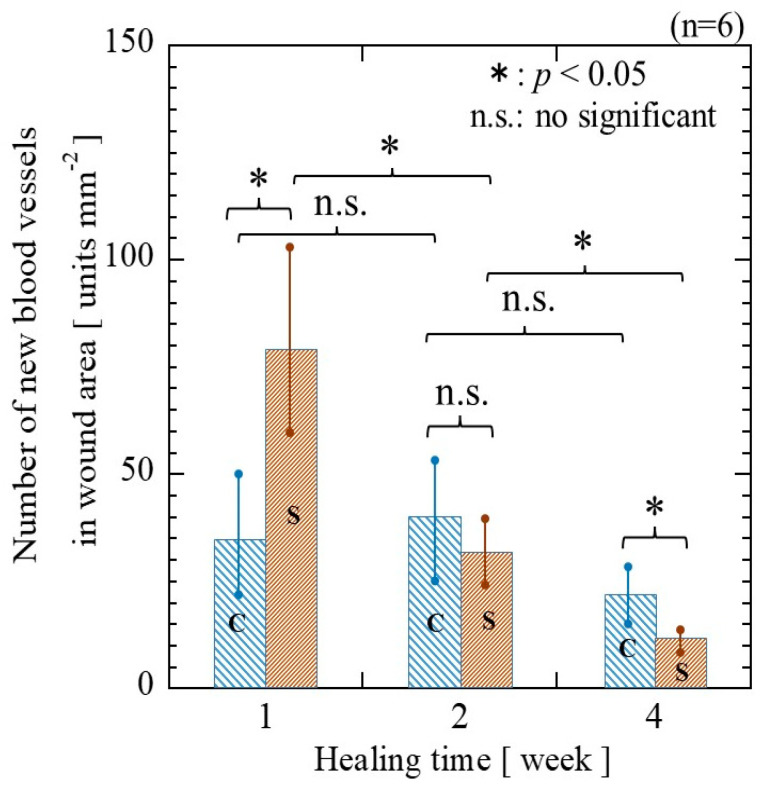
Graph depicting the number of newly formed blood vessels on the wound site after 1, 2, and 4 weeks. C and S mean control and simonkolleite groups, respectively. * = significant difference (*p* < 0.05) and n.s. = not significant.

**Figure 11 bioengineering-10-00375-f011:**
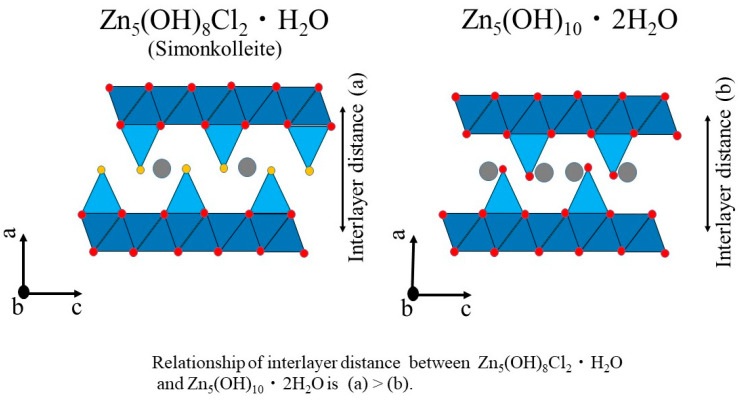
Crystal structures of rhombohedral simonkolleite and monoclinic Zn_5_(OH)_10_·2H_2_O. Stacking layers viewed along the b-direction. The top row shows the layer of edge-sharing octahedra (dark blue) with different arrangements of vacant sites where two tetrahedrally coordinated Zn^2+^ ions (light blue) are situated above and below. Intercalating H_2_O is drawn as gray spheres. Red and yellow spheres represent OH^−^ and Cl^−^, respectively.

**Table 1 bioengineering-10-00375-t001:** Comparison of the proportion of the residual wound between the control and simonkolleite groups after 1, 2, and 4 weeks as determined using the image processing program (%).

Week	0	1	2	4
Control group	100	61 ± 11.5	20 ± 11.3	1.5 ± 0.5
Simonkolleite group	100	52 ± 9.5	12 ± 8.2	0.5 ± 0.5

## Data Availability

No new data were created or analyzed in this study. Data sharing is not applicable to this article.

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
