# Peer review of "The Significant Potential of Simonkolleite Powder for Deep Wound Healing under a Moist Environment: In Vivo Histological Evaluation Using a Rat Model"

_bioengineering, 2023, doi:10.3390/bioengineering10030375_

Round 1

Reviewer 1 Report

For authors:

My sincere thanks go to the authors for their work. Here are my suggestions for improving your work:

1- In the abstract please a) define the “a solution process”??, b) in sentences “the optimal pH (?<pH<?)value for keratinocyte” and below cytotoxic ion concentrations (range ???), please add the optimal pH ranges.  Also, please check the abstract and all text for such improvements that help readerships of the journal and increase your next citations. 

2- Please rearrange keywords as: Ceramics powder; Bioactive ion; Regenerative skin; Wound healing

3- In the graphical abstract please change “Zinc ions sustainable release …..” to sustainable release of Zn2+ ions for wound healing.

4- Please completely revise the last paragraph of the introduction focusing on the novelty.

5- In section 2.1, please add the details of the raw materials such as manufacturer and purity.

6- In section 2.2, please add the details of the SEM (including coating with gold and accelerated voltage) and XRD, including step size and time per step.

7- Please completely re-format the results section. In the result section only, results should be added without discussion! Add the discussion in the discussion section. For example: “Among those elements in the human body, Zn2+ ions have had a big impact on tissue 168 growth and disease progression” and “Sharpe et al. [31] have reported that the optimal pH value for keratinocyte and fibroblast proliferation was 191 in the range of 7.2-8.3” is not related to result section. Move them to the discussion.

8- Change the dimension of ions released to ppm.

9- In XRD results, you should add the used ICDD reference code. Furthermore, I think the XRD graphs of figure 5, do seem not normalized. Please accurately check them and if need normal the data of patterns and then draw them.

10- All microscopic images should be uniform. For example, in figure 7, the scale bar of all figures should locate at the bottom right.

11- Please improve the conclusion section with quantitative data.

12- Suggestion: I think one of the most important issues in the great effect of your powders in wound healing is the antibacterial activity of Zn2+ ions in the wound zones. Please investigate the antibacterial effect of the powders and improve the results and discussion.

Reviewer 2 Report

This manuscript addresses a study evaluating the wound-healing effects of Simonkolleite powder [Zn5(OH)8Cl2・H2O] in rats, demonstrating its ability to accelerate angiogenesis, collagen deposition, and skin deposition tissue regeneration, making it a promising candidate for new wound dressings. Globally this is a good, well-structured, and scientifically sound manuscript and could contribute to various scientific fields involved in wound dressings. My plagiarism check indicated that the content is original and does not match any existing source.

Below I point out some comments and issues that the author should consider for this manuscript:

1.        The TITLE is clear and explanatory but could be enhanced by being more succinct.

2.        The ABSTRACT could benefit from more straightforward language. For example, sentences like "Interestingly, the hair follicles, one of the skin appendages, were observed on the regenerative skin in the simonkolleite group at 4 weeks, but not in the control group" could be simplified and rephrased for better clarity. More specific information about the methods used in the study could be interesting. For instance, knowing how many rats were used in the study and how the simonkolleite powder was applied to the wounds would be helpful. An explicit statement of the study's main conclusion could improve the conclusions. While the sentence "These results suggested that simonkolleite could offer great potential as new wound dressing material" is a good starting point, it could be strengthened by including a summary of the specific ways in which simonkolleite outperformed other wound dressings.

3.        Abbreviations should be checked in the whole document to see if they have been spelled out first or explained;

4.        The INTRODUCTION is well-written and provides a clear overview of the topic and focus of the study, but there are a few areas where it could be improved to provide more context and detail for the reader: (i) a statement that clearly states the hypothesis of the study; (ii) provide more detailed information on the current state of research in healing and skin transplantation; (iii) the current introduction briefly mentions the use of a rat model, it could be helpful to provide more information or the rationale.

5.        The METHODS section could benefit from some improvements, starting by explaining why the methods were chosen. Moreover, in "Synthesis of simonkolleite powder": specify the temperature at which the solutions were prepared; Provide the volume of distilled water used for washing the synthesized powder. Mention the type of ball milling used for grinding the simonkolleite powder. "Powder characterizations": It would be helpful to briefly explain why SEM and XRD were chosen as the analysis methods for the simonkolleite powder. "Zn2+ ions releasing test and pH value": provide a brief explanation of the significance of the 7-day time frame for measuring Zn2+ ion release and pH values; more information on the physiological saline solution used, such as its composition and purpose.

Moreover, in the animal experiments, some improvements could also be considered: provide more detail on the anesthesia protocol used, including the dose and route of administration; the description of the surgical procedure could be made more explicit (it is not clear how the trephine bar was used to create the skin defects); the method for blood vessel quantification could be explained in more detail (how the specific areas for counting were chosen and how the individual counts were combined to calculate the final result).

In the statistical analysis, knowing what specific tests were used to analyze the data and how significance was determined would be helpful.

6.        Overall, the results section provides clear and concise information about the experiments and their outcomes, but minor improvements could be made to improve clarity and readability. For instance, some sentences are long and complex, making the passage difficult to read and understand. For example, the sentence "On the other hand, in the control group, inadequate collagen maturation and residual inflammatory cell infiltration were observed at the same time, which suggests an intermediate stage between the inflammatory and proliferative phases" could be broken down into shorter, simpler sentences to improve clarity. Moreover, when discussing the role of Zn2+ ions in promoting angiogenesis, it would be helpful to provide specific studies showing this effect.

7.        The conclusion section could be improved; for instance: first sentence is long and could be broken up into two sentences for easier readability. When stating that simonkolleite is effective in wound healing, it would be beneficial to quantify this effect by providing statistical data or comparing wound healing rates between the simonkolleite group and the control group. The conclusion could also profit from a short discussion of the implications of the study's findings for wound care and potential future research directions.

The authors have done good laboratory work, and the manuscript corresponds to it.

Reviewer 3 Report

This article, the significant potential of simonkolleite powder for deep wound healing under a moist environment: in vivo histological evaluation using a rat model, very interesting for readers. However, it is will be better if the authors put some relevant reference of your proposed chemical reaction such as eq.3. 

Looking at figure 3, why the surface of particles completely different when compared before and after milled. In fact, the surface morphology affected to their surface area to volume and the healing performance. Did you investigate this parameter? 

The capture of figure 9 and 11 too long. Should be more shorten. Also, sometime use "Fig." and "Figure". It is should be consistent throughout the manuscript.

Round 2

Reviewer 1 Report

Well response.